# Effect of Yield Strength Distribution Welded Joint on Crack Propagation Path and Crack Mechanical Tip Field

**DOI:** 10.3390/ma14174947

**Published:** 2021-08-30

**Authors:** Yueqi Bi, Xiaoming Yuan, Jishuang Lv, Rehmat Bashir, Shuai Wang, He Xue

**Affiliations:** 1National Engineering Laboratory for Coal Mining Machinery and Equipment, Taiyuan Research Institute of China Coal Science and Engineering Group Co., Ltd., Taiyuan 030000, China; 2Taiyuan Research Institute of China Coal Science and Engineering Group Co., Ltd., Taiyuan 030000, China; yuanxiaoming@tyccri.com (X.Y.); mkyljs@126.com (J.L.); 3School of Mechanical and Electrical Engineering, China University of Mining and Technology, Xuzhou 221008, China; 4School of Mechanical Engineering, Xi’an University of Science and Technology, Xi’an 710054, China; rehmatbashir@uet.edu.pk (R.B.); 17101016005@stu.xust.edu.cn (S.W.); xuehe@xust.edu.cn (H.X.); 5Department of Mechanical Engineering, University of Engineering and Technology, Lahore 54890, Pakistan

**Keywords:** welded joint, crack propagation, USDFLD subroutine, XFEM

## Abstract

Due to the particularity of welding processes, the mechanical properties of welded joint materials, especially the yield strength, are unevenly distributed, and there are also a large number of micro cracks, which seriously affects the safety performance of welded joints. In this study, to analyze the effect of the uneven distribution of yield strength on the crack propagation path of welded joints, other mechanical properties and residual stresses of welded joints are ignored. In the ABAQUS 6.14 finite element software, the user-defined field (USDFLD) subroutine is used to define the unevenly distributed yield strength, and extended finite element (XFEM) is used to simulate crack propagation. In addition, the static crack finite element model of the welded joint model is established according to the crack propagation path, which is given the static crack model constant stress intensity factor load, and the influence of an uneven yield strength distribution on mechanical field is analyzed. The results show that the crack length of welded joints as well as the plastic deformation range of the crack tip in high stress areas can be reduced with the increase of yield strength along the crack propagation direction. Moreover, the crack deflects to the low yield strength side. This study provides an analytical reference for the crack path prediction of welded joints.

## 1. Introduction

Material properties have a vital role for the selection of material and the life of a component. During the welding process of welded joints, the material properties of different components are significantly changed by temperature variations, resulting in small cracks [1,2,3]. Under the action of an external load, these small cracks would try to expand, resulting in the failure of the welded joints. Usually, welded joints are regarded as the weakest part of a welded structure [4,5]. The stress–strain condition around the crack is the key factor affecting crack propagation behavior [6]. To express the mechanical heterogeneity of the welded joint is of great significance for calculating and predicting the crack propagation direction and the stress and strain field at the crack tip, considering the safety evaluation of the welded joint. Therefore, study on the variations of the properties due to heterogeneity at the time of the welding process is an important research topic for integrity analysis, and the uneven distribution of local mechanical properties needs to be characterized accurately in welded joints. Different models have been presented to study this issue. A sandwich composite structure can express the mechanical properties of materials in the welded joint area to a certain extent [7,8,9]. Fan et al. [10,11,12] used this method to evaluate the safety of welded joints, established a finite element model of welded joints with dual material sandwich structure, and analyzed the effects of work hardening, initial crack location, and yield strength mismatch on fracture toughness and the crack propagation path. Xue et al. [13,14] ignored the heat affected zone and fusion zone of welded joints, simplified them into sandwich structure, and gave different homogeneous material properties in different regions. The influence of yield strength mismatch on the stress–strain field at the crack tip and the driving force of crack growth were analyzed to predict the crack growth rate. Zhao et al. [15,16] added a heat affected zone when establishing the finite element model of a sandwich structure-welded joint, analyzed the differences of a stress corrosion cracking mechanical field in a heat affected zone under constant load and constant stress intensity factor load, and discussed the influence of yield strength mismatch on a driving force for crack growth. However, the material properties at the interface of the two materials were different. Under the external load, the stress–strain field of the finite element model of the welded joint was discontinuous at the material interface, which seriously affected the accuracy of the safety evaluation of the welded joint.

In most research, the default weld metal and base metal have the same hardening characteristics [17,18], and the mechanical property mismatch defines the ratio of yield strength of the weld metal to the base metal [19]. Based on the finite element model of welded joints simplified as sandwich structure, the effects of strength mismatch on the stress and strain field at the crack tip [20] and the corresponding fracture toughness [21,22] are studied, ignoring the fusion zone and heat affected zone with the change of yield strength. However, there are few studies on the effects of continuously varying yield strength on the stress and strain field at the crack tip and the crack propagation path. In addition, the serious discontinuity of the mechanical field in the analysis of the sandwich structure process will affect the accuracy of the analysis. To solve the discontinuous problem of the mechanical field of sandwich structure, it is very necessary to establish a finite element model for welded joints with a continuous distribution of yield strength. A user-defined field (USDFLD) subroutine in ABAQUS finite element software can be used to introduce solution dependent material properties. This method can be easily defined as a function of field variables [23]. For example, Kong et al. [24] analyzed the strength of the perforated composite pressing plate by combining the test and finite element method. The Tsai Wu coefficient and the stiffness attenuation of the element were controlled by a USDFLD subroutine. The numerical simulation results were in good agreement with the test. Wang et al. [25] used a USDFLD subroutine to establish a modified damage evolution model and established the relationship between damage evolution and load cycle times. Through this model, the life and damage state of the sample were obtained during the finite element analysis. The model could accurately predict the fatigue life of metal under constant amplitude load. In this study, the finite element models of welded joints with a continuous change of yield strength are established by associating yield strength with geometry space through a USDFLD subroutine, and crack propagation is simulated using the extended finite element method XFEM in ABAQUS 6.14 finite element software. The crack propagation path is extracted, and the static crack finite element model without propagation is established. Adjusting the load and maintaining the stress intensity factors of different crack lengths constant and the influence of uneven yield strength distribution on the stress field and strain field at the crack tip are analyzed. The results of this work can be used to help predict the crack propagation path and to improve the safety performance of welded joints.

## 2. Materials and Methods

### 2.1. Inhomogeneous Material Model

Metal material models are usually isotropic elastoplastic models and have a particularly simple form. Thus, the algebraic equations related to model integration can be easily developed based on a single variable, and the material stiffness matrix can be defined. The process of the equation for the inhomogeneous field of the material derivation, which is used to establish USDFLD subroutine, is as follows:

The Mises yield function with the associated flow indicates that there is no volumetric plastic strain [26]; since the elastic bulk modulus is quite large, the volume change will be small. Therefore, we can define the volumetric strain as
(1)$$\varepsilon_{vol}=trace(\varepsilon )$$where $\varepsilon_{vol}$ is the volumetric strain, and ε is the strain matrix.

Therefore, the deviatoric strain e is
(2)$$e=\varepsilon -\frac{1}{3}\varepsilon_{vol}I$$

Strain rate dε can be decomposed into elastic strain rate$d\varepsilon^{el}$ and plastic strain rate $d\varepsilon^{pl}$:(3)$$d\varepsilon =d\varepsilon^{el}+d\varepsilon^{pl}$$

Using the standard definition of corotation measure, this can be written in an integrated form(4)$$\varepsilon =\varepsilon^{el}+\varepsilon^{pl}$$

If elasticity is linear and isotropic, it can be written in terms of two dependent field variables of the material parameters. For this development, it is appropriate to choose these parameters as the bulk modulus *K* and shear modulus *G*. These are readily computer from Young’s modulus E and Poisson’s ratio μ:(5)$$K=\frac{E}{3(1-2\mu )}$$
(6)$$G=\frac{E}{2(1+\mu )}$$

The relationship between volumetric strain and deviatoric stress can be expressed by the bulk modulus:(7)$$p=-K\varepsilon_{vol}$$

Here, p is the isostatic equivalent pressure stress:(8)$$p=-\frac{1}{3}trace(\sigma )$$

The form of the deviatoric matrix is
(9)$$S=2Ge^{el}$$

In Equation (9), **S** is the deviatoric stress matrix:(10)S=σ+pI

Here, I is the identity matrix. The flow rule is described as
(11)$$de^{pl}=de^{-pl}n$$

Here, **n** is the material flow direction:(12)$$n=\frac{3}{2}\frac{S}{q}$$

In Equation (12), *q* is the equivalent yield stress:(13)$$q=\sqrt{\frac{3}{2}S:S}$$
whereas *de^−pl^/dt* is the (scalar) equivalent plastic strain rate. Plasticity requires materials that satisfy uniaxial stress, plastic strain, and strain rate. If the material is rate-independent, Equation (14) is the yield condition:(14)$$q=\sigma^{0}$$
where $\sigma^{0}(\varepsilon^{-pl},\psi )$ is defined by the user as a function of equivalent plastic strain $\varepsilon^{-pl}$ and field variables ψ. The field variable ψ is a function related to space:(15)ψ=f(x,y)

### 2.2. XFEM Theoretical Basis

The crack tip will penetrate the element during the calculation process, and a level set is used to represent the crack position [27]. In XFEM, the displacement function of the crack Gauss point *x* in the element can be expressed by Equation (16) [28]:(16)$$\begin{array}{c} u_{xfem}(x)=\sum_{j=1}^{n}N_{j}(x)u_{j}+\sum_{h=1}^{m_{h}}N_{h}(x)(H(x)-H(x_{h}))a_{h}\\ +\sum_{k=1}^{m_{t}}N_{k}(x)[ \sum_{l=1}^{4}(F_{L}(x)-F_{l}(x_{k}))b_{k}^{l}] \end{array}$$

Here, *n* represents the size of the number of nodes, $N_{j}(x)$ represents the shape function, $u_{j}$ represents the degree of freedom vector of the node, H(x) represents the size of the Heaviside function, $F_{l}(x)$ represents the size of the Gaussian point value at the crack surface, $a_{h}$ is the enhanced node degree of freedom vector on both sides of the crack surface, $m_{h}$ and $m_{l}$ represent the number of reinforcement nodes on both sides of the crack and the number of nodes at the crack tip, and $b_{k}^{l}$ represents the degree of freedom vector, as shown in Figure 1.

The Heaviside enhancement function takes values 1 and −1 on both sides. *F_l_* is the enhancement function in the calculation, and its expression is given in Equation (17) [29]:(17)$${\left\{F_{l}(r,\theta )\right\}}_{l=1}^{4}=\left\{\sqrt{r}\sin\frac{\theta }{2},\sqrt{r}\cos\frac{\theta }{2},\sqrt{r}\sin\theta \sin\frac{\theta }{2},\sqrt{r}\sin\theta \cos\frac{\theta }{2}\right\}$$

Here, (*r*, *θ*) are the coordinates of the Gauss point *x* in the polar coordinate system of the crack tip.

## 3. Numerical Simulation Procedures

### 3.1. Analysis Steps

In general, the micro cracks in welded joints are randomly distributed, and the distribution of the material yield strength is complex. To simplify the model, perpendicular and parallel to the crack are selected as the change direction of yield strength, and the edge-opened crack is selected as the research object. XFEM is used to calculate the crack propagation path. According to the crack propagation path calculated by XFEM, the crack is drawn on the geometry, and the static crack model I is established. When the crack propagates for a certain distance, the crack is updated on the geometry, and static crack model II is established. The crack length of model I is different from that of model II, and the load is also different, but the stress intensity factor *K_I_* calculated according to the load and crack length is the same. The reasons are as follows:

For most engineering materials, there is a plastic zone at the crack tip, as shown in Figure 2. The origin of polar coordinates is the crack tip, the horizontal axis is the polar diameter, and the vertical axis is the tensile stress in the crack tip region along the direction of the zero polar angle. *σ_ys_* is the yield strength of the ideal elastic-plastic material.

In the plastic zone, the stress–strain state is completely different from the linear elastic solution. However, if the size of the plastic zone is sufficiently small and is completely surrounded by the dominant region controlled by *K* field, the stress–strain field in the plastic zone will be controlled by the *K* field. In other words, the geometry, loading mode, and crack size of the two samples are different, but if the stress intensity factors of the two samples are equal, the stress–strain field near the crack tip is also the same.

The red solid line in Figure 2 is Equation (18) [30], which is for the tensile stress of the crack tip along the *r*-axis.
(18)$$\sigma_{yy}\approx \frac{K_{eff}}{\sqrt{2\pi (r-r_{p})}}$$

Due to the existence of the plastic zone, the dotted line around the actual crack is the equivalent crack with the forward propagation length of *r_y_*. The equivalent stress intensity factor is *K_eff_*. When the plastic zone is sufficiently small and when the mode of the crack is the opening crack, *K_eff_* is approximately equal to the stress intensity factor *K**_Ι_*.

The equation of the plastic zone area is:(19)$$r_{p}=\frac{1}{\pi }{\left(\frac{K_{I}}{\sigma_{ys}}\right)}^{2}$$

Here, *r_p_* is the plastic zone length in the *r*-direction, as shown in Figure 2, and *K_Ι_* is the stress intensity factor of the opening crack. σ*_ys_* is the yield stress.

To analyze the distribution range of the stress and strain at the crack tip under the path calculated by XFEM, this study is divided into three analysis steps, as shown in Figure 3.

Step 1: Establish XFEM models with different yield strength distributions and analyze the crack propagation path under constant load *P*_0_, as shown in Figure 3a. Step 2: When the crack length calculated by the XFEM model in Step 1 is a_1_, extract the crack propagation path and establish static crack model I. Step 3: When the crack length in Step 1 is a_2_, extract the crack propagation path and establish static crack model II. The load calculation equation of the static crack model in Step 2 and Step 3 is:(20)$$P=\frac{K_{I}}{F\sqrt{2\pi a}}$$

Here, *P* is the uniformly distributed load, *F* is the correction factor, which is related to the specimen geometry, a is the crack length, and *K_Ι_* is the stress intensity factor.

The load of the static crack model is adjusted to keep the stress intensity factors of all models consistent so that the influence of load on the stress field at the crack tip can be eliminated, and the influence of the uneven distribution of the yield strength on the stress–strain field at the crack tip can be analyzed.

Figure 4 shows the finite element model of the welded joint. Figure 4 shows the finite element model of the XFEM welded joint. The width of the XFEM welded joint is W = 10 mm, and the height is H = 15 mm. The initial crack length is a_0_ = 2 mm, the origin of the coordinate system is at the edge of the crack and the welded joint, and the *x*-direction coincides with the direction of crack propagation. Figure 4a shows that the yield strength changes in the *x*-direction. When the yield strength gradient factor is *G_x_* > 0, the yield strength increases. When the yield strength gradient factor is *G_x_* < 0, the yield strength decreases. Figure 4b shows that the yield strength changes in the *y*-direction. When the yield strength gradient factor is *G_y_* < 0, the yield strength increases. Conversely, when *G_y_* < 0, the yield strength decreases.

Figure 5a shows the finite element model of the XFEM welded joint. The bottom of the welded joint adopts a fixed constraint, and the upper part of the welded joint is a tensile load of 600 MPa. Figure 5b is a partial view of the XFEM model.

Figure 5c is a finite element model of a static cracked welded joint. The geometric dimensions and yield strength distribution of the static crack model are the same as those in Figure 4. The width is W = 10 mm, and the height is H = 15 mm. However, the shape and length of the initial crack is obtained according to the crack propagation path of the XFEM model. Different static crack models apply loads with the constant stress intensity factor *K_Ι_* = 438.66 MPa∙mm^1/2^ to observe variations in the material yield strength gradient on the mechanical field around the crack tip. Figure 5d is a partial view of the crack tip, with the crack tip as the center and radius R = 0.25 mm, which is used to show the characteristics of the mechanical field of the crack tip.

### 3.2. Material Model

Figure 6c shows a tensile test specimen of 306 stainless steel. The size of the sample is shown in Figure 6b. During the test, the pin passes through the holes at both ends of the sample and is hinged with the testing machine. The stress–strain curve of the sample is tested on the tensile testing machine. The uniaxial tensile stress–strain curve of the sample is the engineering stress–strain in Figure 7. Due to shrinkage of the specimen cross section, which will cause the stress–strain curve to be lower, the converted true stress–strain curve is obtained using the ABAQUS 6.14 finite element analysis.

Since the stress–strain curve of 306 stainless steel has no obvious yield point, the stress value at 0.2% of the strain is the yield strength, so the yield strength of the 306 stainless steel is noted to be 400 MPa. Fitting the elastic section of the curve with a straight line, the elastic modulus is 210 GPa. According to the elastic deformation of the sample during the stretching process by the strain gauge, Poisson’s ratio μ is calculated to be 0.3. When the inhomogeneous material field is established in the finite element model, the initial yield strength is 400 MPa, and the yield strength gradient and direction change differently.

In this study, regarding the local range of the welded joints, the change rule of the material yield strength is simplified to a linear one, so the *σ*_0_ is:(21)$$\sigma_{0}=\frac{\Delta \sigma_{s}}{\Delta \psi }(G\psi )+\sigma_{{}_{initial}}$$

*σ_initial_* is the initial yield strength, and $\Delta \sigma_{s}/\Delta \psi$ is the gradient of yield strength, which is a constant in this study. *G* is the yield strength change gradient factor. Since the $\Delta \sigma_{s}/\Delta \psi$ is a constant, the value of *G* indicates the magnitude of the gradient of yield strength change. In the simulation, the yield strength distribution is divided into two groups: group one is the *x*-direction, in which the field variable function is *f*(*x*, *y*) = *x,* and the yield strength gradient factor is recorded as *G_x_*; group two is the *y*-direction, in which the field variable function is *f*(*x*, *y*) = *y,* and the yield strength gradient factor is recorded as *G_y_*. The material properties of the static crack and XFEM model are shown in Table 1 and Table 2.

XFEM crack growth simulation includes three steps: initial crack, crack growth, and failure. All of these steps were performed in ABAQUS 6.14 software. We used the maximum principal stress criterion expressed as Equation (22) [31] for cracks that propagate under tensile load:(22)$$f=\left\{\frac{\langle \sigma_{\max}\rangle }{\sigma_{0\max}}\right\}$$

Here, *f* is the maximum principal stress ratio, $\sigma_{\max}$ is the maximum principal stress. $\sigma_{0\max}$ is the stress at which damage begins, which is set to 600 MPa here. The symbol < > represents the usual interpretation (i.e., $\langle \sigma_{\max}\rangle =0$ if $\sigma_{\max}<0$ and $\langle \sigma_{\max}\rangle =\sigma_{\max}$ if $\sigma_{\max}>0$). Macaulay brackets are used to signify that a purely compressive stress state will not cause damage. Damage is assumed to be when the maximum principal stress ratio reaches 1.

### 3.3. Mesh Generation

Figure 8 shows the meshing of the welded joint finite element model. Figure 8a shows the mesh of the XFEM model. The element type is a 4-node bilinear plane stress quadrilateral reduced integration element (CPS4R), and the number of elements is 13,320. The static crack finite element meshing result is shown in Figure 8b; the element type is also a 4-node bilinear plane stress quadrilateral reduced integration element (CPS4R), and the number of elements is 63,693. The static crack tip local partition is shown in Figure 8c. The red node in Figure 8c is the data extraction node of the stress value and the strain value. The stress curve and the strain curve around the crack tip are sequentially extracted from these nodes. Take the crack as the starting point and follow the clockwise direction as the positive direction of *θ*.

## 4. Results Discussion

### 4.1. The Results of Yield Strength Change in x Direction

A large number of studies have shown that materials with high yield strength have large crack propagation resistance, and materials with low yield strength have low propagation resistance [32,33,34,35]. During the crack propagation process simulated by XFEM, the yield strength of the material through which the crack passes changes, which causes the propagation resistance to change, so the crack length is different.

As shown in Figure 9, the initial crack is parallel to the direction of the material’s yield strength change. The abscissa is the yield strength gradient factor *G_x_*, the ordinate is the crack length a, and the contour shows the material’s yield strength distribution.

When *G_x_* = 0, the yield strength does not change; the material is homogeneous, and the crack length is a*_Gx_*_=0_ = 3.25 mm. When the yield strength gradient factor is *G_x_* > 0, the crack propagates from the low yield strength side to the high yield strength side, and the crack lengths are a*_Gx_*_=0.5_ = 3.16 mm, a*_Gx_*_=1.0_ = 3.08 mm, and a*_Gx_*_=0.5_ > a*_Gx_*_=1.0_. On the contrary, when the yield strength gradient factor is *G_x_* < 0, the crack propagates from the high yield strength side to the low yield strength side, and the crack lengths are a*_Gx_*_=−0.5_ = 3.33 mm, a*_Gx_*_=−1.0_ = 3.42mm, and a*_Gx_*_=−0.5_ < a*_Gx_*_=−1.0_. Comparing the different crack lengths in Figure 7, it can be found that the propagated length of the crack in the material whose yield strength decreases is greater than that whose yield strength increases. That is because the material with low yield strength has low crack propagation resistance, and the crack propagation from high yield strength material to low yield strength material results in a decrease in crack propagation resistance and an increase in crack length. On the contrary, crack propagation from a low yield strength material to a high yield strength material causes the crack growth resistance to increase, and the crack length decreases.

When the yield strength changes in the *x*-direction, the crack tip mechanical field of the crack length at a_1_ = 2 mm is shown in Figure 10. Figure 10a is the Von Mises stress contour at the crack tip, and Figure 10b is the Von Mises stress value of the node set (as shown in Figure 10c). When *G_x_* < 0, Figure 10c,d describes the equivalent plastic strain. The equivalent plastic strain represents the plastic stage of the material. When the material gradually changes from high yield strength to low yield strength, the crack propagates to the low yield strength side. As the yield strength gradient factor *G_x_* of the material decreases, the Von Mises stress field distribution area as well as the Von Mises stress value along the node set at the crack tip is also reduced, as shown in Figure 10a,b. On the contrary, the equivalent plastic strain distribution area in Figure 10c as well as the equivalent plastic strain of the node set in Figure 10d becomes larger.

Figure 10e–h shows the crack tip mechanical field at *G_x_* > 0. The crack propagates to the high yield strength side gradually. With the increase of *G_x_*, the Von Mises stress field distribution area as well as the Von Mises stress value of the crack tip gradually increases, as shown in Figure 10e,f. However, the equivalent plastic strain distribution area in Figure 10g and the curve of equivalent plastic strain in Figure 10h are decreased.

Figure 11 and Figure 12 show the Von Mises stress of the node set along the *x*-axis. The solid line in the figure is the Von Mises stress when the crack length is a_1_ = 2 mm, and the dashed line represents the Von Mises stress of the crack length at a_2_ = 3.0 mm. This shows the crack has expanded under the constant stress intensity factor *K* load. This means that the crack tip mechanical field excludes the influence of the load, and the distribution range of the crack tip mechanical field is determined by the material itself. The Von Mises stress is in a low state at the position where the material has cracked, which is caused by the partial stress release because of the destruction. The stress rises sharply as it approaches the crack tip and reaches the peak point. Mises stress concentrates near the crack tip and forms a high-stress zone. The stress decreases as the distance from the crack tip increases. When it reaches the yield point, the stress decrease rate goes up significantly. When the material is away from the high-stress area near the crack tip, the rate of stress decline slows down, and the material stress is at a lower level. As shown in Figure 11 and Figure 12, when *G_x_* = 0, the material yield strength is uniformly distributed. under the constant stress intensity factor *K* load, the peak stress of different crack lengths is the same, 573.9 MPa. The mechanical field of the crack tip is shown in Figure 13. The crack length does not affect the Von Mises stress field and the equivalent plastic strain distribution area.

In Figure 11, when *G_x_* < 0, the yield strength decreases as the distance D increases. When *G_x_* = −1.0, with the increase in crack length, the peak value of Von Mises stress decreases from 546.9 MPa to 530.9 MPa. Additionally, the peak value of the Von Mises stress decreases from 559.7 MPa to 550.7 MPa at *G_x_* = −0.5. The smaller the value of *G_x_*, the greater the decrease in yield strength, and the greater the decrease in the peak Von Mises stress. Figure 14a,c shows that the Von Mises stress field area decreases with the increase of the crack length. The equivalent plastic strain area of Figure 14b,d increases with the crack length. The larger the increment is, the greater the range of materials entering the plastic stage. At the same time, the Von Mises stress field area in Figure 14a has a larger area reduction than that in Figure 14c. On the contrary, the reduction of the equivalent plastic strain field area in Figure 14b compared to that in Figure 14d is smaller.

When *G_x_* > 0, the yield strength of the material goes up as the distance increases, as shown in Figure 10. With the crack extending, the peak Von Mises stress at the crack tip goes up. When *G_x_* = 1.0, the peak Von Mises stress increases from 603.8 MPa to 619.5 MPa immediately. At this time, the peak Von Mises stress increases from 588.6 MPa to 595.6 MPa at *G_x_* = 0.5. The greater the crack length, the greater the increase in the peak value of the Von Mises stress. The Mises stress field area of the crack tip shown in Figure 14e,g increases with the crack propagation. The equivalent plastic strain area of Figure 14f,h decreases with the crack propagation. The material of the crack tip shows less range in the plastic stage. At the same time, the Mises stress field area in Figure 14e has a larger area increase compared to Figure 14g, and a smaller area decrease in the equivalent plastic strain field in Figure 14f compared to that in Figure 14h.

### 4.2. The Results of Yield Strength Change in y Direction

Figure 15 shows the crack propagation path when the crack is perpendicular to the yield strength change. In Figure 15, the ordinate is the projection length of the crack in the *x*-direction, and the abscissa is the yield strength gradient. The angle between the tangent to the crack tip and the initial crack direction *β* is used to indicate the crack deflection angle. The clockwise deflection angle of the crack is the positive direction. The contour shows the distribution of the yield strength of the material. The red box in the figure is a locally enlarged view of the crack propagation path. When the yield strength gradient is *G_y_* > 0, the left side of the crack is the low yield strength side, and the right side is the high yield strength side. The crack deflection angles under load are *β_Gy=_*_1.0_ = 5.57° and *β_Gy=_*_0.5_ = 0.74°. The crack lengths are a*_Gy=_*_1.0_ = 6.01 mm and a*_Gy=_*_0.5_ = 4.91 mm. When the yield strength gradient is *G_y_* < 0, the right side of the initial crack is the low yield strength side, and the left side is the high yield strength side. The crack deflection angles are *β_Gy=−_*_1.0_ = −5.57° and *β_Gy=−_*_0.5_ = −0.74°, and the crack lengths are a*_Gy=_*_1.0_ = 6.01 mm and a*_Gy=_*_0.5_ = 4.91 mm. When the yield strength gradient is *G_y_* = 0 and when the material is homogeneous, the deflection angle is *β_Gy=_*_0.0_ = 0°; during the crack propagation process, and the crack length is a*_Gy=_*_0.0_ = 4.66 mm. By comparing the relationship between the crack deflection angle and the crack length and the yield strength gradient factor, it can be found that the crack growth process always deflects to the low yield strength side, and the greater value of the absolute value of *G_y_*, the greater the crack deflection angle, and crack length tends to deflect to the low yield strength material.

Figure 16 shows the distribution characteristics of the mechanical field at the crack tip when the yield strength changes in the *y*-direction. When *G_y_* < 0, the material gradually changes from high yield strength to low yield strength in the *y*-direction. Figure 16a shows the Von Mises stress field, and the equivalent plastic strain field is shown in Figure 16c, which depicts deflection in the negative direction of *y*. The Von Mises stress and equivalent plastic strain plots translate to the left, as shown in Figure 16b,d, which indicate that the mechanical field turns to the low yield strength side. As *G_y_* decreases, the deflection of the mechanical field is greater toward the low yield strength side.

On the contrary, when *G_y_* > 0, the material gradually changes from high yield strength to low yield strength in the *y*-direction. The mechanical fields turn to the low yield strength side. As *G_y_* increases, the deflection of the mechanical field is greater toward the low yield strength side.

Regardless of *G_y_* < 0 or *G_y_* > 0, the Mises stress field distribution area of the crack tip is shown in Figure 17a,c,e,g, which decreases with the increase of the crack length. However, the equivalent plastic strain distribution area becomes larger as the crack propagates, as shown in Figure 17b,d,f,h. The crack tip material enters the plastic stage, and the range increases. When the yield strength changes in *y*-direction, the crack deflects to the low yield strength side. In the process of crack propagation, the yield strength in the *x*-direction will decrease. That is why the area of plastic zone at the crack tip goes up and why the distribution area of the high stress zone decreases with the increase of crack length. In the *y*-direction, the crack length goes up with the increase of the absolute value of the yield strength gradient coefficient.

As shown in Figure 18, the yield strength gradient *G_y_* = 0 and thus the yield strength of the material is uniformly distributed. Under the constant stress intensity factor *K* load, when the crack length changes from a_1_ = 2.5 mm to a_2_ = 3 mm, the distribution area of the Von Mises stress and equivalent plastic strain remains unchanged.

## 5. Conclusions

When the crack propagates in the nonuniformly distributed yield strength material, it tends to propagate to the material with low yield strength. The crack length increases with the decrease of yield strength.Keep the stress intensity factor constant. When the crack propagates to the low yield strength side, the area of the Von Mises stress decreases, but the area of equivalent plastic strain increases; when the crack propagates to the high yield strength side, the area of the Von Mises stress distribution goes up, but the area of equivalent plastic strain goes down.When the change of yield strength is perpendicular to the crack propagation direction, the Von Mises stress field and the equivalent plastic strain field at the crack tip deflect to the low yield strength side. The greater the absolute value of the yield strength gradient is, the more obvious the deflection degree is.When the yield strength is parallel to the crack propagation direction, the equivalent plastic strain field at the crack tip spread to the low yield strength side. The Von Mises stress field at the crack tip spread to the high yield strength side.

## Figures and Tables

**Figure 1 materials-14-04947-f001:**
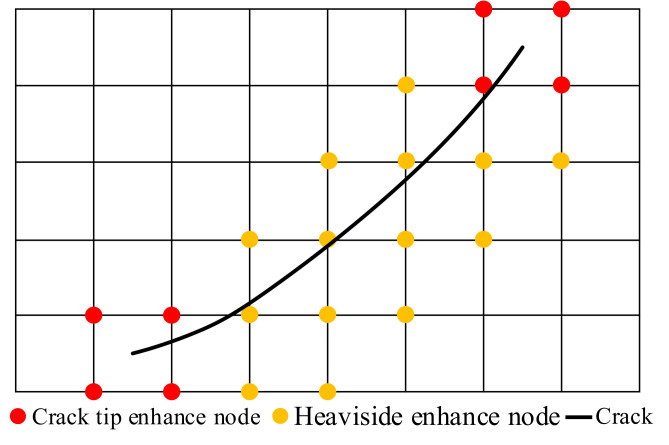
The crack enhancement of XFEM.

**Figure 2 materials-14-04947-f002:**
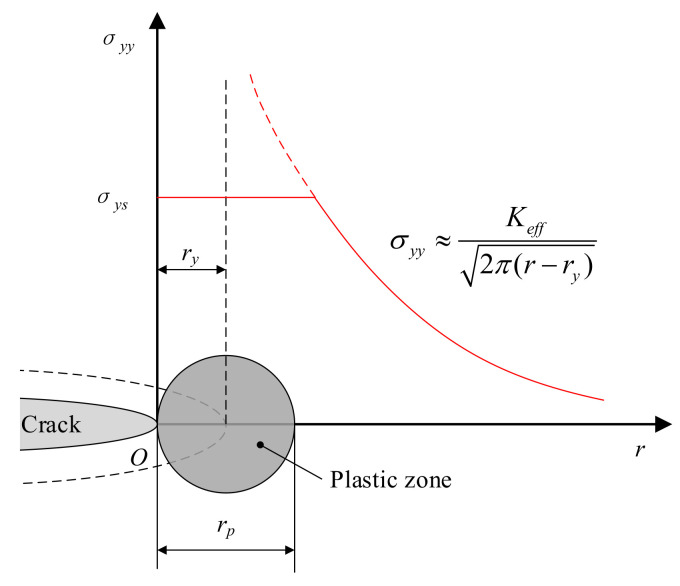
The stress distribution at crack tip of elastic-plastic material.

**Figure 3 materials-14-04947-f003:**
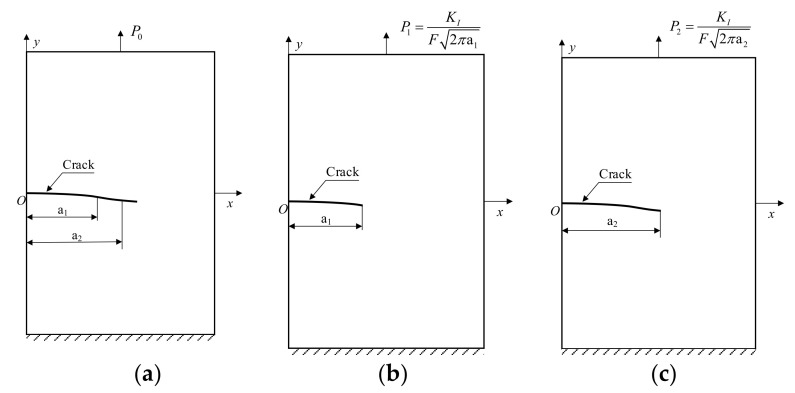
Relationship and analysis steps between (**a**) the XFEM model and (**b**) the static crack model I and (**c**) the static crack model II.

**Figure 4 materials-14-04947-f004:**
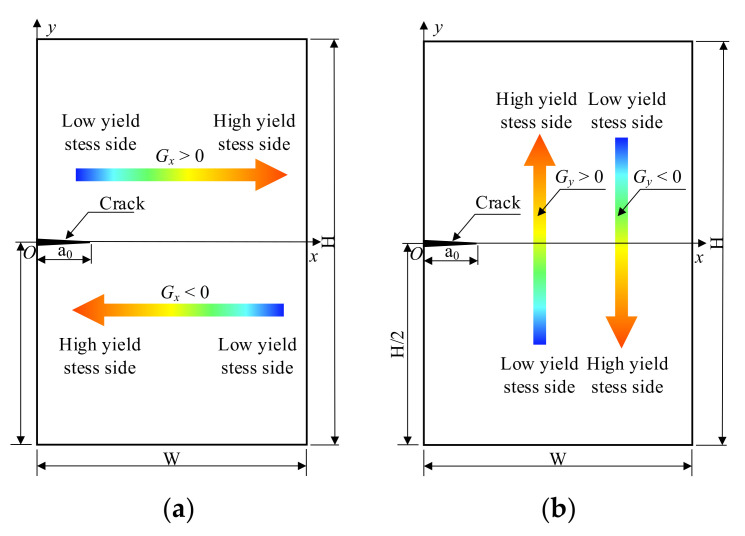
The dimension of material distribution of the welded joint. (**a**) Yield strength changes in *x*-direction and (**b**) in *y*-direction.

**Figure 5 materials-14-04947-f005:**
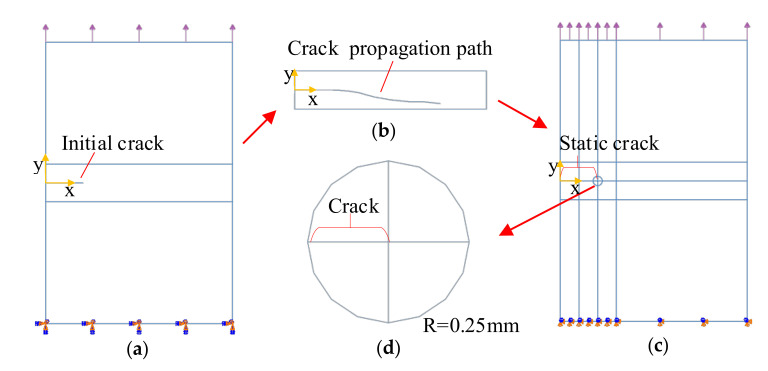
Finite element model of welded joint. (**a**) XFEM model, (**b**) the local view of crack propagation path, (**c**) static crack model, and (**d**) the local view of crack tip.

**Figure 6 materials-14-04947-f006:**
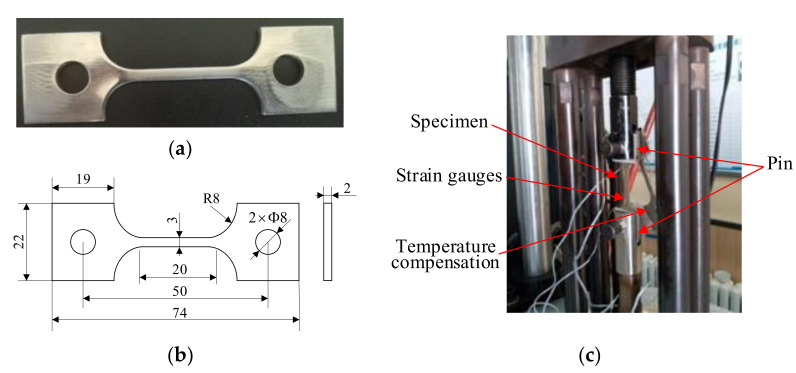
Material tensile test. (**a**) Tensile specimen, (**b**) the size of tensile specimen and (**c**) clamping of tensile specimen.

**Figure 7 materials-14-04947-f007:**
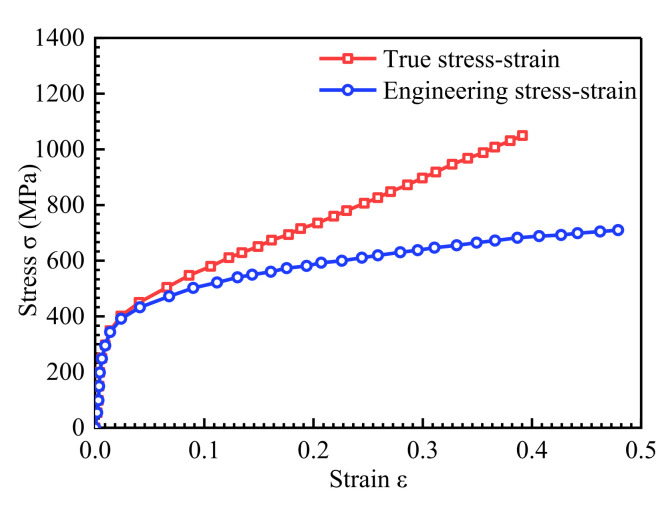
Stress–strain curve of 306 stainless steel.

**Figure 8 materials-14-04947-f008:**
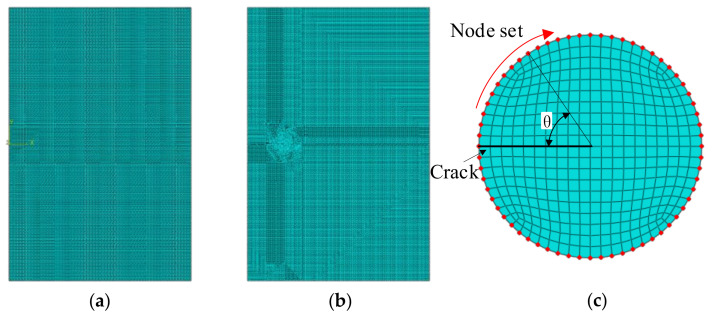
Finite element meshing. (**a**) XFEM model mesh generation, (**b**) Static crack model mesh generation, and (**c**) local mesh generation and node set at crack tip.

**Figure 9 materials-14-04947-f009:**
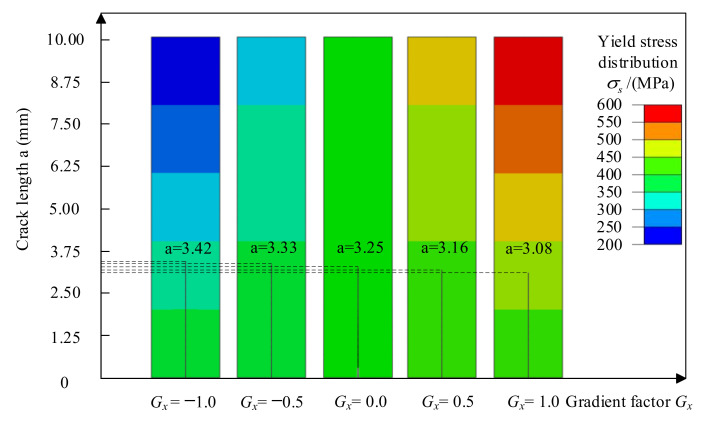
The crack propagation path of the initial crack is parallel to the direction of yield strength change.

**Figure 10 materials-14-04947-f010:**
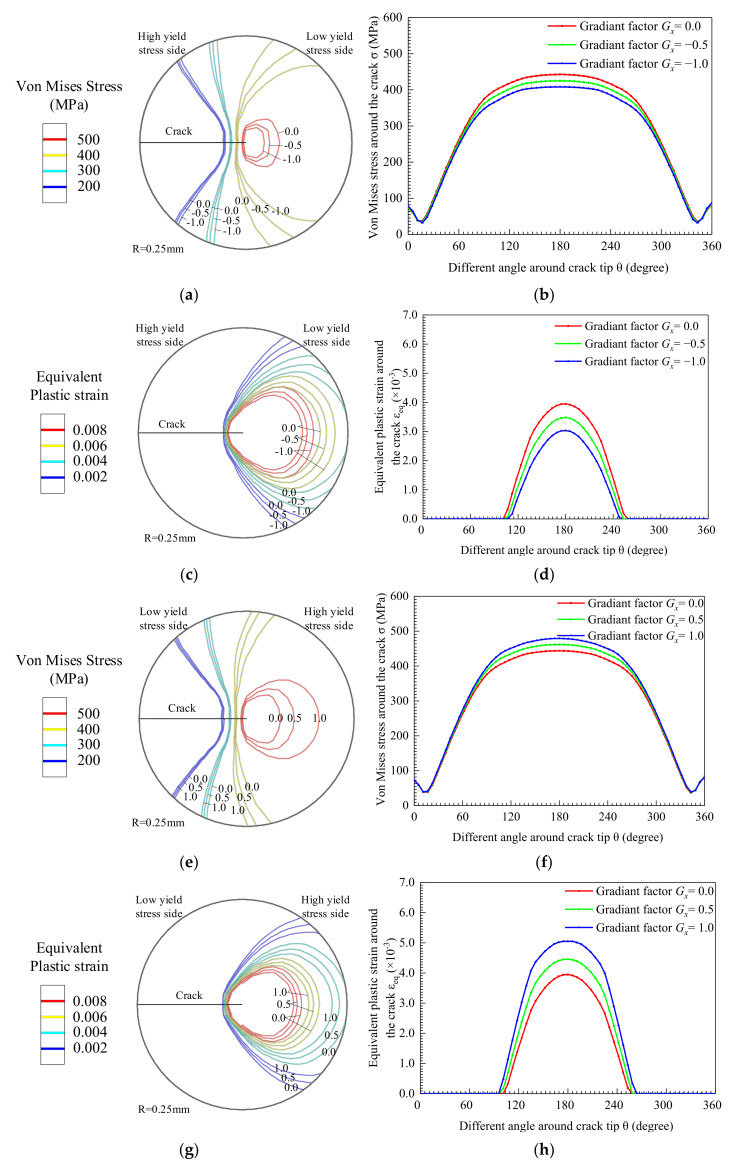
Crack tip mechanical field with crack parallel to the direction of yield strength change. (**a**) *G_x_* < 0 Von Mises stress contour, (**b**) Von Mises stress around crack tip, (**c**) Equivalent plastic strain contour and (**d**) Equivalent plastic strain around crack tip. (**e**) *G_x_* > 0 Von Mises stress contour, (**f**) Von Mises stress around crack tip, (**g**) Equivalent plastic strain contour and (**h**) Equivalent plastic strain around crack tip.

**Figure 11 materials-14-04947-f011:**
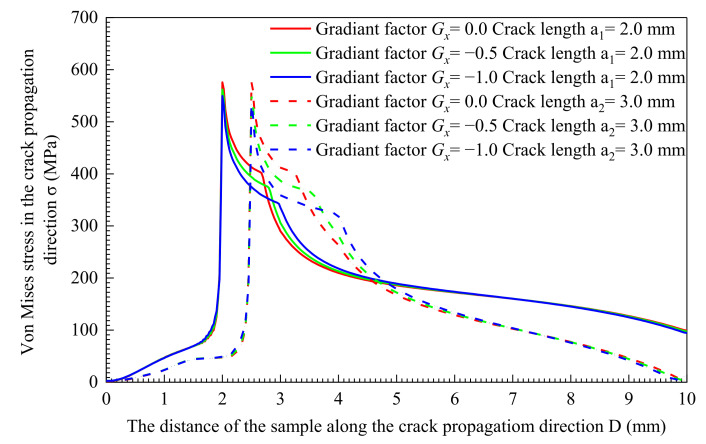
*G_x_* < 0 Von Mises stress distribution characteristics of materials along the crack propagation direction.

**Figure 12 materials-14-04947-f012:**
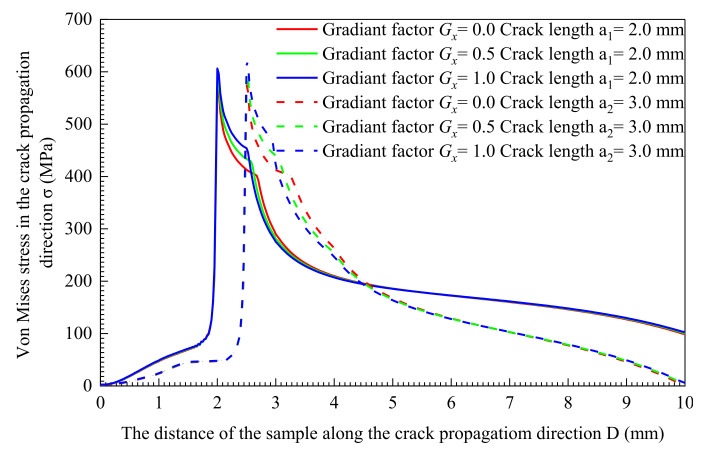
*G_x_* > 0 Von Mises stress distribution characteristics of materials along the crack propagation direction.

**Figure 13 materials-14-04947-f013:**
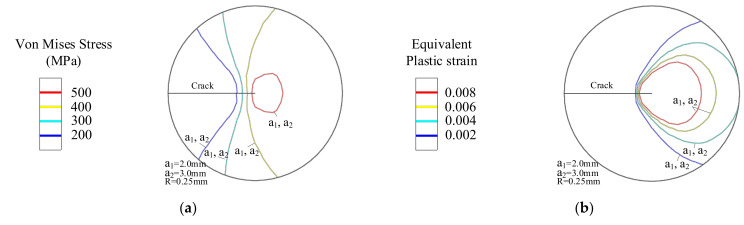
*G_x_* = 0 crack tip mechanical field. (**a**) Von Mises stress contour and (**b**) Equivalent plastic strain contour.

**Figure 14 materials-14-04947-f014:**
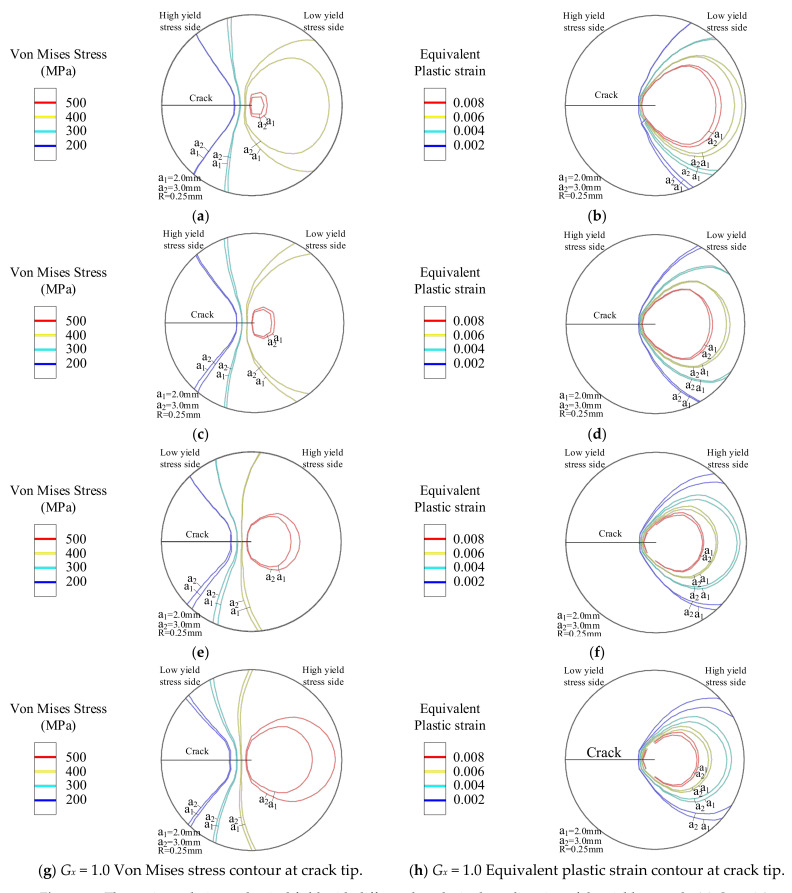
The static crack tip mechanical field with different lengths in the *x*-direction of the yield strength. (**a**) *G_x_* = −1.0 Von Mises stress and (**b**) Equivalent plastic strain contour. (**c**) *G_x_* = −0.5 Von Mises stress and (**d**) Equivalent plastic strain contour. (**e**) *G_x_* = 0.5 Von Mises stress and (**f**) Equivalent plastic strain contour. (**g**) *G_x_* = 1.0 Von Mises stress and (**h**) *G_x_* = 1.0 Equivalent plastic strain contour.

**Figure 15 materials-14-04947-f015:**
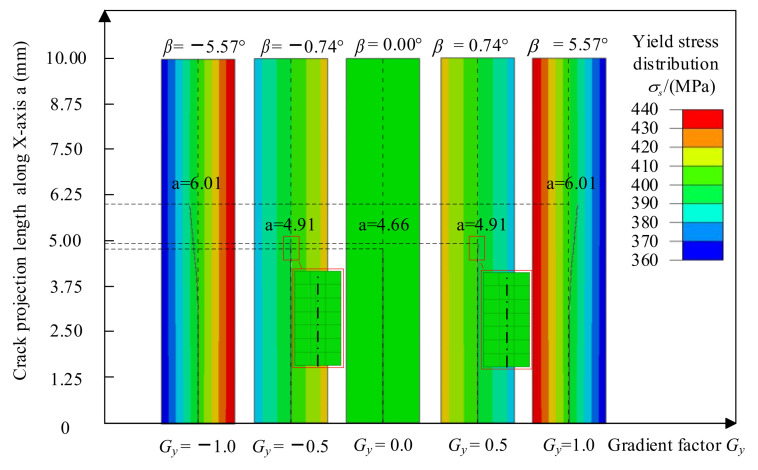
Crack propagation path in the direction of vertical crack yield strength change.

**Figure 16 materials-14-04947-f016:**
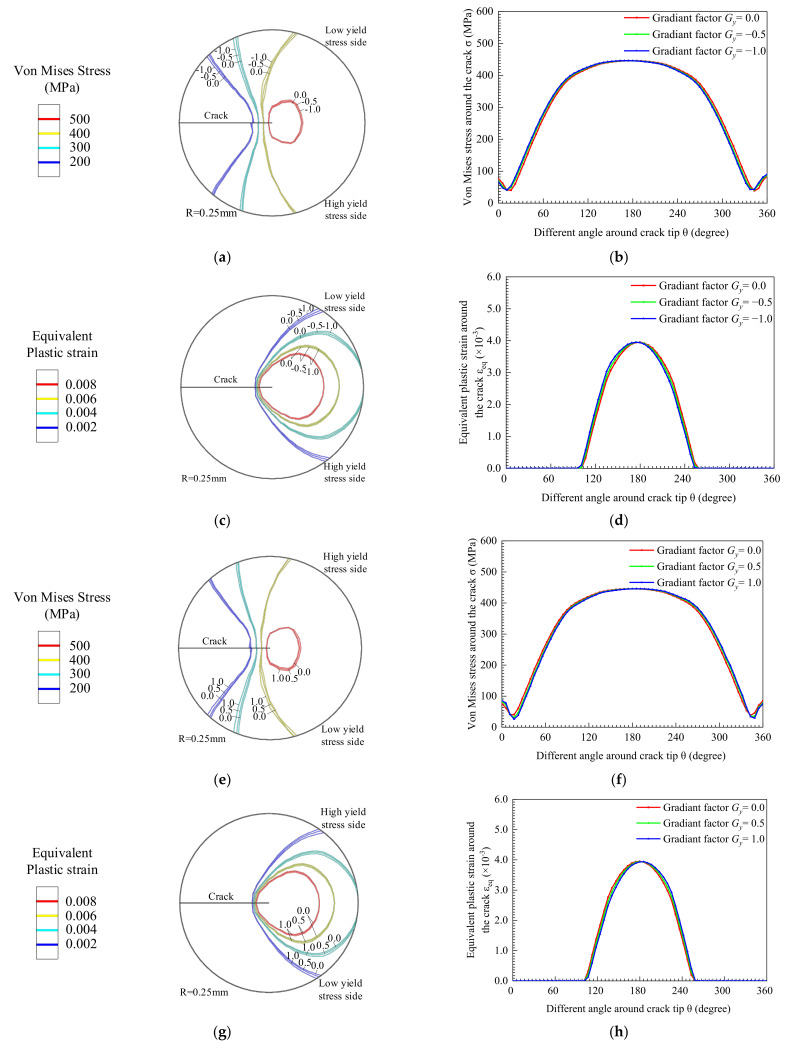
Mechanical field of the crack tip at the initial crack perpendicular to the direction of yield strength change. (**a**) *G_y_* < 0 Von Mises stress contour, (**b**) Von Mises stress around crack tip, (**c**) Equivalent plastic strain contour and (**d**) Equivalent plastic strain around crack tip. (**e**) *G_y_* > 0 Von Mises stress contour, (**f**) Von Mises stress around crack tip, (**g**) Equivalent plastic strain contour and (**h**) Equivalent plastic strain around crack tip.

**Figure 17 materials-14-04947-f017:**
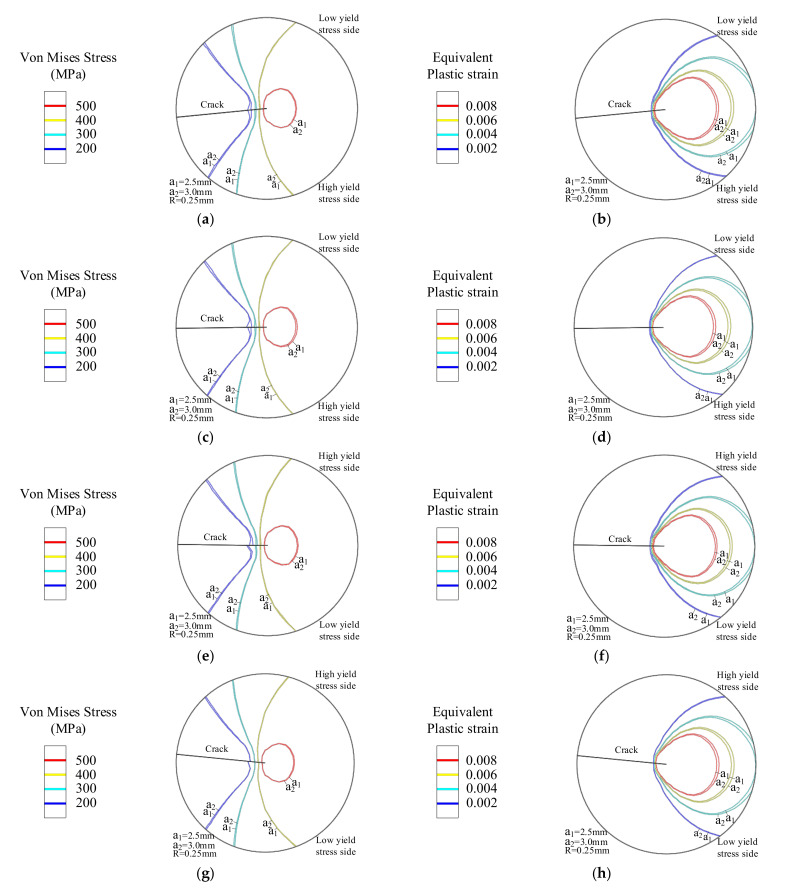
The static crack tip mechanical field with different lengths in the *y*-direction of the yield strength. (**a**) *G_y_* = −1.0 Von Mises stress and (**b**) Equivalent plastic strain contour. (**c**) *G_y_* = −0.5 Von Mises stress and (**d**) Equivalent plastic strain contour. (**e**) *G_y_* = 0.5 Von Mises stress and (**f**) Equivalent plastic strain contour. (**g**) *G_y_* = 1.0 Von Mises stress and (**h**) Equivalent plastic strain contour.

**Figure 18 materials-14-04947-f018:**
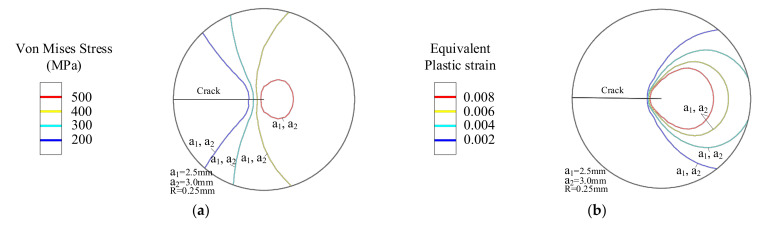
*G_y_* = 0 crack tip mechanical field. (**a**) Von Mises stress contour and (**b**) Equivalent plastic strain contour.

**Table 1 materials-14-04947-t001:** Yield strength changes along the *x*-direction elastoplastic material field.

Group	Elastic Modulus E (GPa)	Poisson’s Ratio μ	Yield Strength Range $\Delta \sigma_{s}/\Delta \psi$ (MPa/mm)	Yield Strength Gradient Factor $G_{x}$
Group 1: Yield strength changes along the *x* direction	210	0.3	26.6	−1.0
				−0.5
				0.0
				0.5
				1.0

**Table 2 materials-14-04947-t002:** Yield strength changes along the *y*-direction elastoplastic material field.

Group	Elastic Modulus E (GPa)	Poisson’s Ratio μ	Yield Strength Range $\Delta \sigma_{s}/\Delta \psi$ (MPa/mm)	Yield Strength Gradient Factor $G_{x}$
Group 2: Yield strength changes along the *y* direction	210	0.3	40.0	−1.0
				−0.5
				0.0
				0.5
				1.0

## Data Availability

The data presented in this study are available on request from the corresponding author.

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
