# Peer review of "Effect of Yield Strength Distribution Welded Joint on Crack Propagation Path and Crack Mechanical Tip Field"

_materials, 2021, doi:10.3390/ma14174947_

Round 1

Reviewer 1 Report

General comments:

  1. Introduction has to be improved, not only by replacing wrongly cited ref. 10, but more by better describing all of them, especially in relevance to the paper itself. Even more important, you have to explain is the purpose of your research? I see no practical aspects of it.
  2. Research design is strange. There is no practical meaning in using such a distribution of yield strengths, welded joints are not designed like that! 
  3. Methods are not well explained, I am still not 100% sure what you actually did by using XFEM and then by static calculation, because it seems that you used XFEM also for static case!
  4. What is meant by constant K load? If both remote stress and crack length are constant, how can you analyse crack propagation?

Author Response

Dear reviewer,
Thank you very much for your careful examination of this paper and your valuable suggestions. Your suggestions are of great help to improve the quality of this paper. I have made modifications according to your suggestions. All the contents are in the first part of the question and reply, and the revised paper is also in the appendix. Please see the attachment. Thank you again for your guidance and help in this paper.
We are looking forward to your reply

Yours sincerely

Yueqi Bi

Reviewer 2 Report

Dear authors,

This is a very interesting paper on "the effect of the non-uniform distribution of the yield strength of welded joints on the crack propagation path and the mechanical tip field of cracks when using USDFLD and XFEM", but for the following reasons, I have to judge as "reject".

[1] Some variables in the formula are not defined.
[2] There are many places where the figure numbers are incorrect.
[3] There are places where the results in the figure and the results in the manuscript do not correspond. (For example, Fig. 13 and the corresponding manuscript)
[4] Please read "Instructions for Authors" of MDPI "Materials" carefully and correct the manuscript according to the writing procedure.
[5] Considering the above, we judge that the quality of the treatise is not in place.

The reference comments for correction are described below.

[1] Title: It is better not to use abbreviations in the title of the treatise.

[2] Introduction "This method is considered to be very suitable for crack growth rate [5]". What is very suitable for expressing the crack growth rate? 

[3] Introduction: "The researchers are working on the prediction of the remaining age and structural integrity evaluation of welded joints". Whether the lifespan here is for static loads or dynamic loads needs to be clearly shown.

[4] Introduction "Wangwei Bing et al. [13] studied that the yield strength mismatch sampling position of nuclear power materials will significantly affect the static and growing crack tip.". What does the term "static" mean?

[5] Equation (2): Variable "I" is not defined.

[6] Equation (3): Strain rate should generally be expressed as ε dots or dε/dt.

[7] Equation (13): I don't understand the expression of Equation (13).

[8] Figure 1: There is no caption in Figure 1.

[9] Equation (17): Please, express the variable on the left side of Equation (17) with r instead of γ (gamma).

[10] Figure 2: etc .: Please lay out the figure after the first appearance in the text.

[11] Figure 2: Please, draw dimensions such as the width and height of the welded joints in the figure. This is because the reader can not imagine the size.

[12] Chapter 2: Describe the dimensions of the tensile test piece and the tensile test conditions in the text.

[13] Chapter 2: In general, 0.2% proof stress is often used as the yield strength for materials whose yield point is unclear. Why did you choose 2% proof stress? Would you please describe the reason in the text?

[14] Chapter 2: I feel that the elastic modulus is generally expressed in GPa. The same applies to Table 1 and Table 2. )

[15] Table 1 and 2: Generally, Poisson's ratio of metal materials is about 0.3, but please explain in the text why the Poisson's ratio used in Tables 1 and 2 was set to 0.25.

[16] Equation (19): σ0max in Equation (19) is not defined in the text.

[17] Chapter 2: I don't understand the meaning of "value path". Would you please explain in the text?

[18] Figure 7: The color scale values ​​of the yield stress distribution in Fig. 7 are displayed in an overlapping manner. Please revise.

[19] Chapter 3: What does "K load" mean? I don't understand the meaning because it suddenly appeared in the text.

[20] Figures 9 and 10: Please, explain in the text how to read the yield stress from Figures 9 and 10.

[21] Figure 13: The values ​​shown in the figure in Fig. 13 do not match the values ​​in the text.

[22] Figure 13: The color scale values ​​of the yield stress distribution in Fig. 13 overlap.

[23] Figure 16: How can we read the yield stress from this figure? Would you please explain in the text?

[24] Conclusion: Please correct the conclusion according to the above points modified.

[25] Others: Other points were written directly in the manuscript. Please refer to it when making corrections.

Best regard,

Author Response

Dear reviewer,

Thank you very much for your careful examination of this paper and your valuable suggestions. Your suggestions are of great help to improve the quality of this paper. I have made modifications according to your suggestions. All the contents are in the second part of the question and reply, and the revised paper is also in the appendix. Please see the attachment. Thank you again for your guidance and help in this paper.

We are looking forward to your reply

Yours sincerely

Yueqi Bi

Reviewer 3 Report

  1. Figure 7 shows that the crack length propagation is parallel to the yield strength direction. What is the width of the crack according to yield strength and its changes needs to be addressed.
  2. On page 13, line 2, please give the figure number.
  3. Figure 1 title is missing.
  4. Some of the equations need a reference since the fundamental equations were used to develop the model.
  5. The technical strength of the discussion needs to be improved by adding some more references.

Author Response

Dear reviewer,

Thank you very much for your careful examination of this paper and your valuable suggestions. Your suggestions are of great help to improve the quality of this paper. I have made modifications according to your suggestions. All the contents are in the third part of the question and reply, and the revised paper is also in the appendix. Please see the attachment. Thank you again for your guidance and help in this paper.

We are looking forward to your reply

Yours sincerely

Yueqi Bi

Round 2

Reviewer 1 Report

Paper is almost ready for publishing, just one more clarification in the introduction, as marked in the attached file.

Author Response

Dear reviewer:

We are very grateful for your valuable suggestions for this paper. Your suggestions will greatly improve the quality of our papers and guide our future research. We have studied comments point by point, revised the manuscript accordingly. The amendments are highlighted in yellow highlight background in the revised manuscript. All authors have approved the response letter and the revised version of the manuscript. Please see the attachment. Thank you again for your guidance and help in this paper.

We are looking forward to your reply.

Yours sincerely!

Yueqi Bi

Reviewer 2 Report

Dear author,

This is a very interesting paper on "Effects of uneven distribution of yield strength of welded joints on crack propagation path and mechanical tip field of cracks when using USDFLD and XFEM", but please consider again following on comments.

[1] Paper title: The term also "XFEM" should not be abbreviated in the paper title.

[2] Equation (13): What does the symbol ":" in Equation (13) mean?

[3] References: Regarding the format of the references, check and correct the references in "Instructions for Authors" again.
https://www.mdpi.com/journal/materials/instructions

Best regard,

Author Response

(The authors gave the same response as above.)

Reviewer 3 Report

I appreciate the author's efforts in the revised manuscript to improve its quality. 

Author Response

Dear reviewer:

Thank you for your careful check and precious suggestions on this paper. It is important for our future research. Once again, we are very grateful for your support for this paper.

Yours sincerely!

Yueqi Bi